# Substituent-Dependent Divergent Synthesis of 2-(3-Amino-2,4-dicyanophenyl)pyrroles, Pyrrolyldienols and 3-Amino-1-acylethylidene-2-cyanopyrrolizines via Reaction of Acylethynylpyrroles with Malononitrile

**DOI:** 10.3390/molecules27238528

**Published:** 2022-12-03

**Authors:** Maxim D. Gotsko, Ivan V. Saliy, Igor A. Ushakov, Lyubov N. Sobenina, Boris A. Trofimov

**Affiliations:** A.E. Favorsky Irkutsk Institute of Chemistry, Siberian Branch, Russian Academy of Sciences, 1 Favorsky Str., 664033 Irkutsk, Russia

**Keywords:** acylethynylpyrroles, malononitrile, 2-(3-amino-2,4-dicyanophenyl)pyrroles, pyrrolyldienols, 3-amino-1-acylethylidene-2-cyanopyrrolizines

## Abstract

An efficient method for the synthesis of pharmaceutically and high-tech prospective 2-(3-amino-2,4-dicyanophenyl)pyrroles (in up to 88% yield) via the reaction of easily available substituted acylethynylpyrroles with malononitrile has been developed. The reaction proceeds in the KOH/MeCN system at 0 °C for 2 h. In the case of 2-acylethynylpyrroles without substituents in the pyrrole ring, the reaction changes direction: instead of the target 2-(3-amino-2,4-dicyanophenyl)pyrroles, the unexpected formation of pyrrolyldienols and products of their intramolecular cyclization, 3-amino-1-acylethylidene-2-cyanopyrrolizines, is observed.

## 1. Introduction8b 5t

Substituted pyrroles are of special significance as key structural units of many natural products [1,2,3,4], pharmacologically active compounds [5,6,7,8], including drugs based on them [9,10,11,12,13], and high-tech materials [14,15,16,17,18]. Among them, of particular importance are aryl-substituted pyrroles, having diverse practically useful properties [19,20,21,22,23,24,25,26,27]. For instance, many pharmaceuticals, such as anti-hyperlipidemic atorvastatin [9,10], potassium-competitive acid blocker vonoprazan [28], marine-derived natural product with anti-tumor activity neolamellarin [29], fluorescent anion sensors [30,31], BODIPY dyes [32,33,34,35,36,37] and ligands for transition metals [38] were designed based on arylpyrroles. Certain arylpyrrole derivatives are intermediates in the synthesis of heterocyclic compounds such as 5-aryl-1,2-dihydro-1-pyrrolizinones with potential anti-inflammatory and analgesic activities [39] or analogs of anticancer drug Mitomycin C, arylpyrrolizines [40].

The recent syntheses of arylpyrroles are based on the introduction of aryl moieties in the pyrrole ring [41,42,43,44,45,46,47] (for instance, the Suzuki–Miyaura coupling [44,47]) and construction of the pyrrole cycle from acyclic compounds with aryl substituents [48,49,50,51]. The reactions of alkylarylketoximes with acetylene [52] or its precursor, calcium carbide [50], which allow the obtainment of various 2-arylpyrroles, are considered to be of a general character. However, even these reactions do not enable the synthesis of pyrroles with some aryl substituents, (e.g., containing both amino and nitrile groups in the benzene ring, in particular, 2,6-dicyanoaniline). This is, probably, due to the inaccessibility of the starting reagents.

At the same time, the compounds containing 2,6-dicyanoaniline scaffold exhibit various biological activities [53,54,55], strong fluorescence [55,56,57,58,59] and may be used as nonlinear optical materials [55,60,61] and cell imaging agents [55,62] (Figure 1). Furthermore, they serve as building blocks for the synthesis of diverse biologically active compounds and a large number of heterocyclic derivatives, including indoles, quinazolines, fluorenones, indazoles, benzoxazines and benzotriazinones [55].

In the light of the foregoing, the synthesis of arylpyrroles with amino and cyano substituents from readily available starting compounds is an important challenge.

Diverse methods for the preparation of 2,6-dicyanoanilines are summarized in review [55].

However, only some of them allow 3,5-disubstituted 2,6-dicyanoanilines with hetocyclic (except for pyrrole) substituents to be synthesized (Figure 1) [55]. These are base-catalyzed reactions of 2-arylidenemalononitrile with 2-(1-arylethylidene)malononitrile (Figure 1a) and ethylenic ketones with malononitrile (Figure 1b).

Our attention was drawn to the methods based on the cyclization of acetylenic ketones with malononitrile, although they did not give 2,6-dicyanoanilines with heterocyclic substituents [56,63,64]. However, these methods have several shortcomings, such as low reaction selectivity or non-mild reaction conditions (reflux in toluene) and the large span of the yields (16–70%). Later [65], the conditions for the selective formation of 3,5-diaryl-2,6-dicyanoanilines via the same reaction with a yield of up to 87% were found (KOH/MeCN, 40 °C, 6 h) (Figure 1c). However, the substrate scope of this reaction is limited only to aryl and thienyl substituents. Perhaps this is due to the inaccessibility of acetylenic ketones with other substituents, such as, in particular, acylethynylpyrrolylketones.

Recently, thanks to the discovery and development of room temperature reaction of pyrroles with electrophilic haloacetylenes in the medium of solid metal oxides and salts, acetylenic ketones with pyrrole and aryl or hetaryl substituents in acyl moiety have become readily available [66,67,68,69,70] and widely used as a rewarding platform for design of polyfunctional pyrrole compounds [71,72].

## 2. Results and Discussion

In the present paper, we describe the reaction of acylethynylpyrroles **1a–p**, synthesized according to Figure 2, with malononitrile **2**. The study has been undertaken in order to develop an effective method for the synthesis of pyrroles with a 2,6-dicyanoaniline substituent.

According to our preliminary experiments, under the abovementioned conditions [65] (KOH/MeCN, 40 °C, 6 h), the reaction of 1-methyl-2-(benzoylethynyl)pyrrole **1a** with malononitrile **2** (the ratio **1a** : **2** : KOH is 1 : 2 : 2) did not afford the expected product, the process was accompanied by resinification. Therefore, in order to find suitable conditions for the construction of 1-methyl-2-(3-amino-2,4-dicyanophenyl)pyrrole **3a**, we further carried out the reaction of acylethynylpyrrole **1a** with malononitrile **2** at room temperature. However, the tarring of the reaction mixture was also observed in this case. It was possible to obtain pyrrole **3a** in 78% yield only at 0 °C for 2 h. It should be emphasized that the reaction is completely selective: no other products except for pyrrole **3a** were found in the reaction mixture.

Since the conversion of the starting acylethynylpyrrole **1a** was complete under the conditions employed, we used them for the reaction of malononitrile with other substituted acylethynylpyrroles **1b**–**1m**. As a result, a number of previously unknown 2-(3-amino-2,4-dicyanophenyl)pyrroles **3b–3m** were synthesized in good to excellent yields (Table 1). 

As follows from Table 1, the reaction is equally effective for acylethynylpyrroles with alkyl and aryl substituents at carbon atoms in the pyrrole ring and also for 4,5,6,7-tetrahydroindole derivatives. The nature of the substituent at the acyl function (aryl or hetaryl), and at the nitrogen atom (H, methyl, benzyl or vinyl) almost does not affect the outcome of the reaction.

To our surprise, in the case of acylethynylpyrroles **1n–p** with unsubstituted pyrrole ring, we encountered quite a different reaction: under the same conditions (KOH/MeCN, the ratio of **1** : **2** : base is 1 : 2 : 2, 0 °C, 2 h), instead of the expected 2-(3-amino-2,4-dicyanophenyl)pyrroles **3**, the adducts of malononitrile and acylethynylpyrrole in the enol form, pyrrolyldienols **4a–c**, were formed. The latter were isolated either in pure form (in the case of enol **4a**) or in the mixtures (in the case of enols **4b,c**) with small amounts (10–14%) of their keto tautomers **5b,c** (Figure 3).

It should be emphasized that, according to the ^1^H NMR data, dienols **4a–c** and ketones **5a–c** are Z-isomers, the latter being stabilized by intramolecular hydrogen bond between the NH-proton and C = O group (δ NH—15.4 15.9 ppm).

During isolation and drying of the crude product (especially at elevated temperature), the mixture of tautomers **4b,c** and **5b,c** was selectively transformed to aminopyrrolizines **6b,c** in 55 and 64% yields, respectively. The corresponding enol **4a** turned out to be stable and was isolated in 75% yield without impurity of keto-tautomer and the corresponding pyrrolizine. Aminopyrrolizine **6a** was prepared in 80% yield by refluxing enol **4a** in ethanol in the presence of triethylamine (Figure 4).

The formation of aminopyrrolizines obviously °C curs as intramolecular addition of the NH pyrrole moiety to the cyano group followed by prototropic isomerization of the imino-group to NH_2_ substituent (Figure 3). The formation of aminopyrrolizines is strictly stereoselective: the *Z*-isomers are formed exclusively with *cis*-position of the proton at the double bond and the ortho-proton of aryl group, which follows from the main NOESY (

) and HMBC (

) correlations in 2D NMR spectra (Figure 2). 2M NMR spectra of aminopyrrolizine 6c confirm its the Z-form.

Thus, the substituent-dependent divergent synthesis of three different products has been achieved: in the case of acylethynylpyrroles with substituents in the pyrrole ring, 2-(3-amino-2,4-dicyanophenyl)pyrroles **3** are formed, while for substrate with unsubstituted pyrrole ring, pyrrolyldienols **4** and their keto-tautomers **5** are produced, which latter can be cyclized to aminopyrrolizines **6**.

Apparently, the formation of 2-(3-amino-2,4-dicyanophenyl)pyrroles **3** starts with proton abstraction from the CH_2_-group of malononitrile. The carbanion **A**, thus generated, adds to the triple bond of acylethynylpyrroles **1**, followed by Knoevenagel condensation of the adduct **B**, thus formed, with second molecule of malononitrile. Otherwise, malononitrile first reacts with the carbonyl group, and its second molecule adds to the triple bond of the intermediate pyrrolylenyne **D**. Next, diadduct **C** undergoes intramolecular cyclization with the participation of carbanion **E** to give cyclic imine **F**. Hydrolysis of one of the nitrile groups followed by decarboxylation and aromatization of the intermediate **G** finishes the process (Figure 5).

Our assumption that one of the stages in the assembly of the aniline ring is the addition of acetonitrile to the carbonyl group, as we recently showed on the example of the formation of pyridines from acylethynylpyrroles [73], was not confirmed: when the reaction of acylethynylpyrrole **1d** with malononitrile was carried out without acetonitrile in THF, the yield of pyrrole **3d** was 38%.

In the framework of the above mechanism, the substituent-dependent divergence of the synthesis can be explained as follows. In the presence of KOH, monoadducts of malononitrle to acylethynylpyrroles **1n–p** should exist mainly as pyrrolate-anions with the negative charge distributed over the whole molecules along the conjugated chain up to the carbonyl group, which accepts a part of the negative charge, as shown by the resonance forms in (Figure 6).

Such a charge transfer should be mostly expressed for the substrates with unsubstituted pyrrole counterpart that results in the decreasing of electrophilicity of the carbonyl group, which loses the ability to add the second molecule of malononitrile. Alkyl substituents in the pyrrole ring disfavor the above charge transfer from the pyrrolate-anions due to the reduction of the pyrrole moiety acidity. In the case of 5-arylsubstituted pyrrole structures, the negative charges of the pyrrolates are partially distributed over the aromatic substituents. N-substituted substrates do not form pyrrolate-anions at all. Thus, monoadducts with unsubstituted pyrrole counterparts **1n–p** appear to be incapable of being attacked by the second molecule of malononitrile and hence of affording 2-(3-amino-2,4-dicyanophenyl)pyrroles **3**. Instead, they isomerize to dienols **4a–c**, which in favorable conformation undergo the ring closure to pyrrolizines **6a–c**. On the other hand, monoadducts having any substituents in the pyrrole ring are readily attacked by the second carbanion of malononitrile at their more electrophilic carbonyl group to construct dicyanoaniline structures.

## 3. Materials and Methods

### 3.1. General Information

IR spectra were obtained with a Bruker Vertex 70 spectrometer (400–4000 cm^−1^, KBr). ^1^H (400.13 MHz), ^13^C (100.6 MHz) spectra were recorded on a Bruker DPX-400 spectrometer at ambient temperature in CDCl_3_ solutions and referenced to CDCl_3_ (residual protons of solvent in ^1^H NMR δ = 7.27 ppm; ^13^C NMR δ = 77.1 ppm) and DMSO-d_6_ (residual protons of solvent in ^1^H NMR δ = 2.50 ppm; ^13^C NMR δ = 39.5 ppm). The assignment of signals in the ^1^H NMR spectra was made using COSY and NOESY experiments. Resonance signals of carbon atoms were assigned based on ^1^H- ^13^C HSQC and ^1^H-^13^C HMBC experiments. The C, H, N, S microanalyses were performed on a Flash EA 1112 CHNS-O/MAS analyzer. Clorine was determined by mercurimetric titration. Sulfur was determined by complexometric titration with Chlorasenazo III. Melting point (uncorrected) was determined on a Kofler micro hot-stage apparatus.

Malononitrile **2**, KOH and MeCN are commercial products. Acylethynylpyrroles were prepared according to the procedure reported [66].

### 3.2. Synthesis of 2-(3-amino-2,4-dicyanophenyl)pyrroles (3a–m), pyrrolyldienols (4a–c) and pyrrolizines (6b,c) (General Procedure)

The suspension of malononitrile **2** (132 mg, 2 mmol) and KOH·0.5H_2_O (130 mg, 2 mmol) in acetonitrile (15 mL) was stirred at 20–25 °C for 30 min. Then the reaction mixture was cooled to 0 °C, the 2-acylethynylpyrrole **1** (1 mmol) in acetonitrile (5 mL) was added dropwise to a reaction mixture within 10 min. Reaction mixture was stirred at 0 °C for 2 h and then was diluted with water (40 mL), extracted with diethyl ether (4 × 10 mL). Extracts were washed with water and dried over Na_2_SO_4_. The residue, after removing solvent, was fractionated by column chromatography (Al_2_O_3_, *n-*hexane: diethyl ether, 1 : 1) to afford the pyrroles **3a–m**.

Under analogous conditions, pyrrolyldienols **4a–c** are formed from 2-acylethynylpyrroles **1n–p** and malononitrile **2**. They are isolated by filtering the formed precipitates by diluting the reaction mixtures with water (1:3). Pyrrolyldienols **4b,c** are formed with additive of the keto form. Products **4b,c** were transformed in the corresponding pyrrolizines **6b,c** in the course of isolation and drying.

### 3.3. Characterization Data of Products 3, 4, 6b,c

3-Amino-5-(1-methyl-1*H*-pyrrol-2-yl)-[1,1′-biphenyl]-2,4-dicarbonitrile (**3a**). Yield 233 mg (78%); yellow solid, m.p. 213–215 °C. R_f_ = 0.68. ^1^H NMR (400.13 MHz, DMSO-d_6_): δ 7.62–7.61 (m, 2H, *o*-Ph), 7.54–7.51 (m, 3H, *m,p-*Ph), 7.01–6.98 (m, 1H, H-5, pyrrole), 6.75 (s, 1H, CH, aniline), 6.74 (br. s, 2H, NH_2_), 6.46–6.45 (m, 1H, H-4, pyrrole), 6.17–6.15 (m, 1H, H-3, pyrrole), 3.65 (s, 3H, CH_3_). ^13^C NMR (100.6 MHz, DMSO-d_6_): δ 154.1, 149.3, 140.9, 137.4, 129.3, 128.9, 128.6 (2C), 128.5 (2C), 126.6, 118.5, 116.2, 116.1, 112.2, 107.9, 94.4, 93.2, 34.9. IR (KBr, cm^−1^): 3471, 3356, 3240, 2927, 2853, 2214, 2161, 1640, 1585, 1572, 1550, 1498, 1463, 1419, 1380, 1289, 1143, 1076, 1032, 959, 909, 733, 700, 633, 496. Elemental analysis calculated (%) for C_19_H_14_N_4_: C, 76.49; H, 4.73; N, 18.78; found: C, 76.6; H, 4.86; N, 18.96.

3-Amino-5-(4-ethyl-5-propyl-1*H*-pyrrol-2-yl)-[1,1′-biphenyl]-2,4-dicarbonitrile (**3b**). Yield 266 mg (75%); yellow solid, m.p. 118–120 °C. R_f_ = 0.66. ^1^H NMR (400.13 MHz, CDCl_3_): δ 8.91 (br. s, 1H, NH), 7.56–7.53 (m, 2H, *o*-Ph), 7.49–7.47 (m, 3H, *m,p-*Ph), 6.92 (s, 1H, CH, aniline), 6.85 (d, *J* = 2.8 Hz, 1H, H-3, pyrrole), 5.27 (br. s, 2H, NH_2_), 2.63–2.57 (m, 2H, CH_2_), 2.45 (q, *J* = 7.6 Hz, 2H, CH_2_)_,_ 1.71–1.62 (m, 4H, CH_2_), 1.19 (t, *J* = 7.5 Hz, 3H, CH_3_), 0.99 (t, *J* = 7.3 Hz, 3H, CH_3_). ^13^C NMR (100.6 MHz, CDCl_3_): δ 154.0, 149.7, 139.3, 138.0, 134.6, 129.6, 128.9 (2C), 128.4 (2C), 125.2, 125.0, 118.2, 116.7, 115.5, 113.7, 91.3, 88.0, 28.2, 23.0, 18.9, 15.6, 14.0. IR (KBr, cm^−1^): 3473, 3357, 3239, 2962, 2928, 2871, 2208, 2156, 1959, 1633, 1587, 1571, 1541, 1490, 1459, 1329, 1274, 1208, 1176, 1139, 1075, 1041, 1018, 908, 858, 820, 772, 731, 700, 649, 503. Elemental analysis calculated (%) for C_23_H_22_N_4_: C, 77.94; H, 6.26; N, 15.81; found: C, 77.75; H, 6.40; N, 15.91.

3-Amino-5-(4,5,6,7-tetrahydro-1*H*-indol-2-yl)-[1,1′-biphenyl]-2,4-dicarbonitrile (**3c**). Yield 244 mg (72%); yellow solid, m.p. 175–177 °C. R_f_ = 0.67. ^1^H NMR (400.13 MHz, DMSO-d_6_): δ 11.21 (br. s, 1H, NH), 7.63–7.60 (m, 2H, *o-*Ph), 7.54–7.51 (m, 3H, *m,p-*Ph), 7.01 (s, 1H, CH, aniline), 6.90 (s, 1H, H-3, pyrrole), 6.47 (br. s, 2H, NH_2_)_,_ 2.61–2.57 (m, 2H, CH_2_-7), 2.50–2.46 (m, 2H, CH_2_-4), 1.78–1.74 (m, 2H, CH_2_-5), 1.71–1.64 (m, 2H, CH_2_-6). ^13^C NMR (100.6 MHz, DMSO-d_6_): δ 154.8, 149.1, 139.5, 138.0, 133.0, 129.2, 128.5 (2C), 128.4 (2C), 124.9, 119.0, 117.4, 116.6, 114.0, 111.2, 90.0, 87.9, 23.2, 22.7, 22.4, 22.4. IR (KBr, cm^−1^): 3468, 3348, 3241, 2924, 2852, 2361, 2209, 1637, 1570, 1544, 1489, 1460, 1362, 1270, 1205, 1144, 1126, 1012, 931, 897, 851, 805, 771, 699, 637, 505. Elemental analysis calculated (%) for C_22_H_18_N_4_: C, 78.08; H, 5.36; N, 16.56; found: C, 77.86; H, 5.55; N, 16.38.

3-Amino-5-(5-phenyl-1*H*-pyrrol-2-yl)-[1,1′-biphenyl]-2,4-dicarbonitrile (**3d**). Yield 303 mg (84%); yellow solid, m.p. 82–84 °C. R_f_ = 0.66. ^1^H NMR (400.13 MHz, CDCl_3_): δ 9.49 (br. s, 1H, NH), 7.59–7.40 (m, 10H, Ph), 7.04–7.03 (m, 1H, H-3, pyrrole), 7.02 (s, 1H, CH, aniline), 6.66–6.64 (m, 1H, H-4, pyrrole), 5.31 (br. s, 2H, NH_2_). ^13^C NMR (100.6 MHz, CDCl_3_): δ 153.9, 150.0, 138.9, 137.8, 136.9, 131.3, 129.8, 129.3 (2C), 129.0 (2C), 128.4 (2C), 128.3, 127.8, 124.5 (2C), 118.2, 116.4, 116.1, 115.1, 108.8, 92.4, 88.7. IR (KBr, cm^−1^): 3456, 3357, 3242, 3061, 2923, 2852, 2361, 2253, 2211, 1634, 1577, 1546, 1476, 1460, 1424, 1385, 1298, 1262, 1216, 1077, 1058, 856, 758, 700, 497. Elemental analysis calculated (%) for C_24_H_16_N_4_: C, 79.98; H, 4.47; N, 15.55; found: C, 79.71; H, 4.32; N, 15.63.

3-Amino-5-(5-(4-chlorophenyl)-1*H*-pyrrol-2-yl)-[1,1′-biphenyl]-2,4-dicarbonitrile (**3e**). Yield 316 mg (80%); yellow solid, m.p. 236–238 °C. R_f_ = 0.68. ^1^H NMR (400.13 MHz, DMSO-d_6_): δ 11.64 (br. s, 1H, NH), 7.82–7.80 (m, 2H, Ph), 7.67–7.65 (m, 2H, Ph), 7.58–7.53 (m, 3H, Ph), 7.47–7.44 (m, 2H, Ph), 7.23 (s, 1H, CH, aniline), 7.11–7.10 (m, 1H, H-3, pyrrole), 6.78–6.77 (m, 1H, H-4, pyrrole), 6.63 (br. s, 2H, NH_2_). ^13^C NMR (100.6 MHz, DMSO-d_6_): δ 154.6, 149.4, 139.1, 137.8, 134.7, 131.3, 130.4, 129.3, 128.9, 128.6 (2C), 128.6 (2C), 128.5 (2C), 126.4 (2C), 117.0, 116.3, 115.6, 113.7, 108.9, 91.8, 89.7. IR (KBr, cm^−1^): 3464, 3446, 3353, 3240, 2214, 1645, 1584, 1574, 1541, 1472, 1450, 1368, 1309, 1293, 1256, 1245, 1216, 1098, 1077, 1054, 1011, 933, 819, 767, 751, 696, 639, 607, 529. Elemental analysis calculated (%) for C_24_H_15_ClN_4_: C, 73.00; H, 3.83; Cl, 8.98; N, 14.19; found: C, 72.63; H, 3.64; N, 14.00.

3-Amino-5-(1-vinyl-4,5,6,7-tetrahydro-1*H*-indol-2-yl)-[1,1′-biphenyl]-2,4-dicarbonitrile (**3f**). Yield 321 mg (88%), yellow solid, m.p. 182–184 °C. R_f_ = 0.65. ^1^H NMR (400.13 MHz, CDCl_3_): δ 7.55–7.52 (m, 2H, Ph), 7.50–7.45 (m, 3H, Ph), 6.82 (s, 1H, CH, aniline), 6.73 (dd, *J* = 15.6, 8.9 Hz, 1H, H_x_), 6.51 (s, 1H, H-3, pyrrole), 5.31 (br. s, 2H, NH_2_), 4.95 (d, *J* = 8.9 Hz, 1H, H_b_), 4.88 (d, *J* =15.6 Hz, 1H, 1H_a_), 2.68–2.65 (m, 2H, CH_2_-7), 2.56–2.53 (m, 2H, CH_2_-4), 1.88–1.83 (m, 2H, CH_2_-5), 1.79–1.73 (m, 2H, CH_2_-6). ^13^C NMR (100.6 MHz, CDCl_3_) δ 153.5, 149.1, 141.2, 137.7, 133.7, 130.9, 129.7, 129.0 (2C), 128.5 (2C), 128.4, 120.5, 120.4, 116.6, 116.4, 114.8, 107.1, 94.2, 93.3, 23.9, 23.3, 23.2, 23.1. IR (KBr, cm^−1^): 3471, 3356, 3240, 2927, 2853, 2214, 2161, 1640, 1585, 1572, 1550, 1498, 1463, 1419, 1380, 1289, 1143, 1076, 1032, 959, 909, 733, 700, 633, 496. Elemental analysis calculated (%) for C_24_H_20_N_4_: C, 79.10; H, 5.53; N, 15.37; found: C, 79.21; H, 5.71; N, 15.55.

3-Amino-5-(1-benzyl-4,5,6,7-tetrahydro-1*H*-indol-2-yl)-[1,1′-biphenyl]-2,4-dicarbonitrile (**3g**). Yield 347 mg (81%), yellow solid, m.p. 137–139 °C. R_f_ = 0.67. ^1^H NMR (400.13 MHz, CDCl_3_): δ 7.43–7.28 (m, 7H, Ph), 7.23–7.19 (m, 1H, Ph), 6.87–6.85 (m, 2H, *o*-H, CH_2_Ph), 6.62 (s, 1H, CH, aniline), 6.51 (s, 1H, H-3, pyrrole), 5.29 (br. s, 2H, NH_2_), 5.11 (s, 2H, CH_2_-Ph), 2.61–2.58 (m, 2H, CH_2_-7), 2.45–2.43 (m, 2H, CH_2_-4), 1.83–1.73 (m, 4H, CH_2_-5,6). ^13^C NMR (100.6 MHz, CDCl_3_): δ 153.6, 149.1, 141.3, 138.5, 137.4, 134.2, 129.6, 129.0 (2C), 128.8 (2C), 128.6, 128.3 (2C), 127.3, 125.6 (2C), 119.6, 119.5, 116.6, 116.4, 112.8, 94.6, 93.1, 48.0, 23.5, 23.2, 23.1, 22.5. IR (KBr, cm^−1^): 3467, 3354, 3239, 3061, 3031, 2928, 2850, 2214, 1632, 1568, 1552, 1496, 1457, 1382, 1287, 1118, 1029, 910, 805, 731, 700, 649, 495. Elemental analysis calculated (%) for C_29_H_24_N_4_: C, 81.28; H, 5.65; N, 13.07; found: C, 81.01; H, 5.41; N, 13.24.

2-Amino-4-(1-benzyl-4,5,6,7-tetrahydro-1*H*-indol-2-yl)-6-(furan-2-yl)isophthalonitrile (**3h**). Yield 352 mg (84%), yellow solid, m.p. 215–217 °C. R_f_ = 0.66. ^1^H NMR (400.13 MHz, CDCl_3_): δ 7.41–7.40 (m, 1H, H-5, furan), 7.33–7.29 (m, 2H, *m-*H, CH_2_Ph), 7.25–7.23 (m, 1H, H-3, furan), 7.02–7.01 (m, 1H, *p*-H, CH_2_Ph), 7.00 (s, 1H, CH, aniline), 6.90–6.88 (m, 2H, *o*-H, CH_2_Ph), 6.53 (s, 1H, H-3, pyrrole), 6.47 (dd, *J* = 3.5, 1.7 Hz, 1H, H-4, furan), 5.26 (br. s, 2H, NH_2_), 5.12 (s, 2H, CH_2_-Ph), 2.60 (m, 2H, CH_2_-7), 2.46 (m, 2H, CH_2_-4), 1.73 (m, 4H, CH_2_-5,6). ^13^C NMR (100.6 MHz, CDCl_3_) δ 153.8, 149.3, 144.4, 141.2, 138.6, 135.9, 134.3, 129.0 (2C), 128.6, 127.3, 125.7 (2C), 119.6, 116.7 (2C), 115.1, 112.8, 112.6, 112.5, 93.7, 88.2, 48.0, 23.6, 23.2, 23.2, 22.5. IR (KBr, cm^−1^): 3356, 2923, 2852, 2212, 1723, 1634, 1587, 1558, 1493, 1380, 1294, 1029, 731. Elemental analysis calculated (%) for C_27_H_22_N_4_O: C, 77.49; H, 5.30; N, 13.39; found: C, 77.26; H, 5.45; N, 13.58.

2-Amino-4-(1-benzyl-4,5,6,7-tetrahydro-1*H*-indol-2-yl)-6-(thiophen-2-yl)isophthalonitrile **(3i).** Yield 343 mg (79%), yellow solid, m.p. 197–199 °C. R_f_ = 0.66. ^1^H NMR (400.13 MHz, CDCl_3_): δ 7.51–7.45 (m, 1H, H-5, thiophene), 7.38–7.37 (m, 1H, H-3, thiophene), 7.32–7.21 (m, 3H, Ph, H-4 thiophene), 7.08–7.05 (m, 1H, Ph), 6.90–6.88 (m, 2H, Ph), 6.77 (s, 1H, CH, aniline), 6.53 (s, 1H, H-3, pyrrole), 5.29 (br. s, 2H, NH_2_), 5.11 (s, 2H, CH_2_-Ph), 2.61–2.58 (m, 2H, CH_2_-7), 2.45–2.42 (m, 2H, CH_2_-4), 1.83–1.73 (m, 4H, CH_2_-5,6). ^13^C NMR (100.6 MHz, CDCl_3_): δ 154.0, 141.2, 140.8, 139.0, 138.4, 134.3, 129.1 (2C), 128.6, 128.5, 128.4, 128.2, 127.4, 125.6 (2C), 119.7, 118.3, 116.7, 116.6, 112.9, 94.2, 91.21 48.0, 23.5, 23.2, 23.1, 22.5. IR (KBr, cm^−1^): 3467, 3354, 3239, 2927, 2851, 2211, 2157, 1632, 1566, 1494, 1463, 1437, 1381, 1357, 1289, 1144, 1112, 909, 844, 806, 728. Elemental analysis calculated (%) for C_27_H_22_N_4_S: C, 74.63; H, 5.10; N, 12.89; S, 7.38; found: C, 74.29; H, 5.27; N, 13.03; S, 7.27.

3-Amino-5-(1-methyl-5-phenyl-1*H*-pyrrol-2-yl)-[1,1′-biphenyl]-2,4-dicarbonitrile **(3j).** Yield 326 mg (87%), yellow solid, m.p. 228–230 °C. R_f_ = 0.68. ^1^H NMR (400.13 MHz, CDCl_3_): δ 7.60–7.58 (m, 2H, Ph), 7.54–7.42 (m, 7H, Ph), 7.38–7.34 (m, 1H, Ph), 6.88 (s, 1H, CH, aniline), 6.69 (d, *J* = 3.8 Hz, 1H, H-3, pyrrole), 6.38 (d, *J* = 3.8 Hz, 1H, H-4, pyrrole), 5.39 (br. s, 2H, NH_2_), 3.62 (s, 3H, CH_3_). ^13^C NMR (100.6 MHz, CDCl_3_): δ 153.5, 149.7, 141.1, 140.5, 137.6, 132.6, 131.9, 129.8, 129.2 (2C), 129.1 (2C), 128.7 (2C), 128.5 (2C), 127.8, 120.1, 116.6, 116.3, 113.6, 109.9, 94.8, 94.0, 34.9. IR (KBr, cm^−1^): 3357, 2923, 2854, 2214, 1634, 1576, 1549, 1460, 1285, 758, 728, 700. Elemental analysis calculated (%) for C_25_H_18_N_4_: C, 80.19; H, 4.85; N, 14.96. found: C, 80.03; H, 5.01; N, 15.08.

3-Amino-5-(1-benzyl-5-phenyl-1*H*-pyrrol-2-yl)-[1,1′-biphenyl]-2,4-dicarbonitrile **(3k).** Yield 369 mg (82%); yellow solid, m.p. 150–152 °C. R_f_ = 0.68. ^1^H NMR (400.13 MHz, CDCl_3_): δ 7.47–7.45 (m, 3H, Ph), 7.42–7.33 (m, 7H, Ph), 7.14–7.12 (m, 3H, Ph), 6.77 (s, 1H, CH, aniline), 6.67 (d, *J* = 3.7 Hz, 1H, H-3, pyrrole), 6.63–6.61 (m, 2H, Ph), 6.44 (d, *J* = 3.7 Hz, 1H, H-4, pyrrole), 5.30 (br. s, 2H, NH_2_), 5.24 (s, 2H, CH_2_-Ph). ^13^C NMR (100.6 MHz, CDCl_3_): δ 153.2, 149.5, 141.5, 140.5, 138.5, 137.4, 132.8, 131.7, 129.7, 129.3 (2C), 129.0 (2C), 128.9, 128.7 (2C), 128.6 (2C), 128.4 (2C), 127.9, 127.3, 126.0 (2C), 120.2, 116.2, 114.3, 110.7, 95.6, 94.1, 49.8. IR (KBr, cm^−1^): 3464, 3356, 3062, 3031, 2928, 2216, 1630, 1580, 1550, 1500, 1456, 1389, 1352, 1286, 1181, 1075, 1032, 910, 760, 732, 701. Elemental analysis calculated (%) for C_31_H_22_N_4_: C, 82.64; H, 4.92; N, 12.44; found: C, 82.32; H, 5.02; N, 12.57.

2-Amino-4-(1-benzyl-5-phenyl-1*H*-pyrrol-2-yl)-6-(furan-2-yl)isophthalonitrile **(3l).** Yield 374 mg (85%), yellow solid, m.p. 176–178 °C. R_f_ = 0.67. ^1^H NMR (400.13 MHz, CDCl_3_): δ 7.56–7.54 (m, 1H, H-5, furan), 7.44–7.32 (m, 5H, Ph), 7.27–7.26 (m, 1H, Ph), 7.21 (s, 1H, CH, aniline), 7.14–7.10 (m, 3H, Ph, H-3, furan), 6.70 (d, *J* = 3.7 Hz, 1H, H-3, pyrrole), 6.64–6.62 (m, 2H, Ph), 6.57 (dd, *J* = 3.4, 1.6 Hz, 1H, H-4, furan), 6.45 (d, *J* = 3.7 Hz, 1H, H-4, pyrrole), 5.30–5.23 (m, 4H, NH_2,_
CH_2_-Ph). ^13^C NMR (100.6 MHz, CDCl_3_): δ 153.5, 149.2, 144.6, 141.4, 140.5, 138.5, 136.1, 132.9, 131.7, 129.4 (2C), 128.7 (2C), 128.6 (2C), 127.9, 127.3, 126.0 (2C), 116.6, 116.2, 115.7, 114.3, 112.9, 112.8, 110.8, 94.7, 89.0, 49.7. IR (KBr, cm^−1^): 3357, 2922, 2853, 2213, 1633, 1586, 1554, 1479, 1454, 1294, 1029, 910, 755, 732, 701. Elemental analysis calculated (%) for C_29_H_20_N_4_O: C, 79.07; H, 4.58; N, 12.72; found: C, 78.84; H, 4.35; N, 12.56.

2-Amino-4-(1-benzyl-5-phenyl-1*H*-pyrrol-2-yl)-6-(thiophen-2-yl)isophthalonitrile **(3m).** Yield 402 mg (88%), yellow solid, m.p. 203–205 °C. R_f_ = 0.66. ^1^H NMR (400.13 MHz, CDCl_3_): δ 7.63–7.58 (m, 1H, H-5, thiophene), 7.50–7.47 (m, 1H, H-3, thiophene), 7.41–7.32 (m, 5H, Ph), 7.18–7.12 (m, 4H, Ph, H-4, thiophene), 6.91 (s, 1H, CH, aniline), 6.67–6.65 (m, 1H, H-3, pyrrole), 6.62–6.61 (m, 2H, Ph), 6.43–6.41 (m, 1H, H-3, pyrrole), 5.28 (br. s, 2H, NH_2_), 5.24 (s, 2H, CH_2_-Ph). ^13^C NMR (100.6 MHz, CDCl_3_): δ 153.6, 141.5, 141.2, 140.6, 138.8, 138.4, 132.8, 131.8, 129.4 (2C), 128.7 (2C), 128.7 (4C), 128.5, 127.9, 127.4, 126.0 (2C), 119.3, 116.5, 116.1, 114.3, 110.6, 95.3, 92.2, 49.8. IR (KBr, cm^−1^): 3355, 2921, 2852, 2213, 1632, 1576, 1546, 1458, 1425, 1288, 1254, 910, 845, 759, 731, 702. Elemental analysis calculated (%) for C_29_H_20_N_4_S: C, 76.29; H, 4.42; N, 12.27; S, 7.02; found: C, 76.01; H, 4.32; N, 12.04; S, 6.84.

(*Z*)-2-(3-Hydroxy-3-phenyl-1-(1*H*-pyrrol-2-yl)allylidene)malononitrile **(4a)**. Yield 196 mg (75%), yellow solid, m.p. 175–177 °C. R_f_ = 0.42. ^1^H NMR (400.13 MHz, DMSO-d_6_): δ 12.08 (br. s, 1H, OH), 8.36 (br. s, 1H, NH), 8.00–7.98 (m, 2H, Ph), 7.59–7.57 (m, 3H, Ph), 7.51–7.50 (m, 1H, H-5, pyrrole), 7.44–7.42 (m, 1H, H-4, pyrrole), 7.31 (s, 1H, = CH), 6.46–6.45 (m, 1H, H-3, pyrrole). ^13^C NMR (100.6 MHz, DMSO-d_6_): δ 158.4, 156.1, 144.8, 131.6, 130.6, 129.0 (2C), 126.4, 125.8 (2C), 125.0, 117.6, 116.2, 112.0, 97.6, 85.6. IR (KBr, cm^−1^): 3300, 2216, 1630, 1540, 1500, 1467, 1362, 1325, 1052, 994, 910, 780, 699, 642. Elemental analysis calculated (%) for C_16_H_11_N_3_O: C, 73.55; H, 4.24; N, 16.08; found: C, 73.21; H, 4.42; N, 15.88.

(*Z*)-2-(3-Hydroxy-3-(2-furyl)-1-(1*H*-pyrrol-2-yl)allylidene)malononitrile **(4b)**. ^1^H NMR (400.13 MHz, DMSO-d_6_): δ 12.22 (br. s, 1H, OH), 8.78 (br. s, 1H, NH), 8.06–8.04 (m, 1H, H-5, furan), 7.52–7.51 (m, 1H, H-3, furan), 7.40–7.37 (m, 1H, H-3, pyrrole), 7.16–7.15 (m, 1H, H-4, furan), 7.13 (s, 1H, = CH), 6.81–6.80 (m, 1H, H-5, pyrrole), 6.49–6.46 (m, 1H, H-4, pyrrole).

(*Z*)-2-(3-Hydroxy-1-(1*H*-pyrrol-2-yl)-3-(thiophen-2-yl)allylidene)malononitrile **(4c)**. ^1^H NMR (400.13 MHz, DMSO-d_6_): δ 12.14 (br. s, 1H, OH), 8.81 (br. s, 1H, NH), 7.96–7.95 (m, 1H, H-5, thiophene), 7.84–7.83 (m, 1H, H-3, thiophene), 7.52–7.51 (m, 1H, H-3, pyrrole), 7.47–7.45 (m, 1H, H-5, pyrrole), 7.32–7.30 (m, 1H, H-4, thiophene), 7.24 (s, 1H, = CH), 6.47–6.46 (m, 1H, H-4, pyrrole). ^13^C NMR (100.6 MHz, DMSO-d_6_): δ 157.0, 154.7, 145.3, 133.9, 132.1, 129.2, 129.0, 127.3, 124.9, 117.3, 116.9, 112.3, 97.0, 83.8.

(*Z*)-3-Amino-1-(2-(furan-2-yl)-2-oxoethylidene)-1*H*-pyrrolizine-2-carbonitrile (**6b**). Yield 161 mg (64%), orange solid, m.p. 206–208 °C. R_f_ = 0.36. ^1^H NMR (400.13 MHz, DMSO-d_6_): δ 9.15 (s, 2H, NH_2_), 7.97–7.94 (m, 1H, H-5, furan), 7.53–7.51 (m, 1H, H-5, pyrrole), 7.47–7.44 (m, 1H, H-3, furan), 6.33–6.31 (m, 1H, H-4, furan), 6.72–6.69 (m, 1H, H-3, pyrrole), 6.57 (s, 1H, = CH), 6.48–6.45 (m, 1H, H-4, pyrrole). ^13^C NMR (100.6 MHz, DMSO-d_6_): δ 175.2, 154.9, 154.2, 146.4, 142.0, 131.5, 118.3, 117.1, 116.7, 115.6, 115.5, 112.7, 104.1, 67.2. IR (KBr, cm^−1^): 3134, 2205, 1687, 1622, 1557, 1540, 1492, 1466, 1398, 1260, 1235, 1206, 1114, 1087, 1067, 1039, 1017, 968, 886, 811, 722, 669. Elemental analysis calculated (%) for C_14_H_9_N_3_O_2_: C, 66.93; H, 3.61; N, 16.73; found: C, 67.12; H, 3.80; N, 16.91.

(*Z*)-3-Amino-1-(2-oxo-2-(thiophen-2-yl)ethylidene)-1*H*-pyrrolizine-2-carbonitrile **(6c).** Yield 147 mg (55%), red solid, m.p. 283–285 °C. R_f_ = 0.36. ^1^H NMR (400.13 MHz, DMSO-d_6_): δ 9.15 (s, 2H, NH_2_), 7.91–7.90 (m, 1H, H-5, thiophene), 7.88–7.86 (m, 1H, H-3, thiophene), 7.53–7.51 (m, 1H, H-5, pyrrole), 7.45–7.44 (m, 1H, H-3, pyrrole), 7.25–7.22 (m, 1H, H-4, thiophene), 6.56 (s, 1H, = CH), 6.48–6.45 (m, 1H, H-4, pyrrole). ^13^C NMR (100.6 MHz, DMSO-d_6_): δ 179.1, 155.0, 146.9, 142.0, 133.3, 131.5, 130.4, 128.7, 118.2, 117.2, 116.7, 115.6, 104.3, 67.1. IR (KBr, cm^−1^): 3323, 3153, 2204, 2173, 1629, 1590, 1556, 1509, 1449, 1416, 1382, 1339, 1250, 1152, 1059, 998, 959, 877, 857, 813, 775, 738, 715, 600, 534. Elemental analysis calculated (%) for C_14_H_9_N_3_OS: C, 62.91; H, 3.39; N, 15.72; S, 11.99; found: C, 63.05; H, 3.55; N, 15.91; S, 11.86.

### 3.4. The synthesis of (Z)-3-amino-1-(2-oxo-2-phenylethylidene)-1H-pyrrolizine-2-carbonitrile (6a)

The triethylamine (101 mg, 1 mmol) was added to the solution of pyrrolyldienol **4a** (261 mg, 1 mmol) in EtOH (40 mL) and refluxed for 30 min. After the reaction mixture was cooled to room temperature, the resulting crystalline precipitate **6a** was filtered off on a Schott filter and dried under vacuum. Yield 209 mg (80%), brown solid, m.p. 158–160 °C. R_f_ = 0.35. ^1^H NMR (400.13 MHz, DMSO-d_6_): δ 9.15 (s, 2H, NH_2_), 7.97–7.95 (m, 2H, H-5, H-3, pyrrole), 7.61–7.53 (m, 5H, Ph), 6.68 (m, 1H, H-4, pyrrole), 6.46 (s, 1H, = CH). ^13^C NMR (100.6 MHz, DMSO-d_6_): δ 186.4, 154.9, 142.3, 139.5, 131.9, 131.6, 128.7 (2C), 127.5 (2C), 118.3, 117.1, 116.7, 115.7, 104.5, 67.4. IR (KBr, cm^−1^): 3314, 2206, 1638, 1591, 1512, 1443, 1398, 1366, 1282, 1180, 1147, 1057, 966, 872, 823, 757, 698, 608, 531. Elemental analysis calculated (%) for C_16_H_11_N_3_O: C, 73.55; H, 4.24; N, 16.08; found: C, 73.73; H, 4.38; N, 16.27.

## 4. Conclusions

In summary, an effective access to 2-(3-amino-2,4-dicyanophenyl)pyrroles, pyrrolyldienols and 3-amino-1-acylethylidene-2-cyanopyrrolizines, attractive objects for drug design, via substituent-dependent divergent reaction of acylethynylpyrroles with malononitrile, has been developed. The method has several synthetic advantages, such as the one-pot procedure, very mild conditions (0 °C), the use of readily available starting materials and good-to-excellent yields of the above compounds, and therefore can stimulate the interest of both synthetic and pharmaceutical communities.

## Data Availability

The data presented in this study are available in the Appendix A.

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
