# Peer review of "Substituent-Dependent Divergent Synthesis of 2-(3-Amino-2,4-dicyanophenyl)pyrroles, Pyrrolyldienols and 3-Amino-1-acylethylidene-2-cyanopyrrolizines via Reaction of Acylethynylpyrroles with Malononitrile"

_molecules, 2022, doi:10.3390/molecules27238528_

Round 1

Reviewer 1 Report

The organization and analysis of the reported manuscript were of high quality. I think it can be published in this journal after minor revision.

1: The manuscript has too many sections, such as (in page 3 line 82-88 and some other places), please check and modify.

2: I suggest that the authors to update some references, and quote some latest results on pyrrole-based bioactive natural products and drugs for improved readability and wide readership. For example, references 5-8 can be replaced by doi: 10.1007/s11030-022-10387-8 and doi: 10.2174/1871520622666220613140607.

3: It is suggested to highlight the novelty of this work clearly in introduction. In page 2 line 51-52, “diverse methods for the preparation of 2,6-dicyanoanilines” can be presented in the form of a Table to highlight the novelty of this study. This is critical to address in this manuscript, the authors should enrich this part in the revised version.

4: Please supplement the Rf values of all compounds.  

5: In page 4 line 124 “Figure 1” should be “Figure 2”.

Author Response

Reviewer wrote:

1: The manuscript has too many sections, such as (in page 3 line 82-88 and some other places), please check and modify.

Answer:

Checked as recommended. Sections have combined in some places (see, for instance, page 2, paragraph 2,  page 4, paragraph 2).

2: I suggest that the authors to update some references, and quote some latest results on pyrrole-based bioactive natural products and drugs for improved readability and wide readership. For example, references 5-8 can be replaced by doi: 10.1007/s11030-022-10387-8 and doi: 10.2174/1871520622666220613140607.

Answer:

The works recommended by the reviewer were included in the text as references 7, 8.

3: It is suggested to highlight the novelty of this work clearly in introduction. In page 2 line 51-52, “diverse methods for the preparation of 2,6-dicyanoanilines” can be presented in the form of a Table to highlight the novelty of this study. This is critical to address in this manuscript, the authors should enrich this part in the revised version.

Answer:

Done as recommended.  Diverse methods for the preparation of 3,5-disubstituted 2,6-dicyanoanilines with heterocyclic substituents, which are closest in structure to the compounds synthesized by us, were included into the text as Scheme 1.

4: Please supplement the Rf values of all compounds.  

Answer:

Rf values of all compounds were added in experimental part.

5: In page 4 line 124 “Figure 1” should be “Figure 2”.

Answer:

“Figure 1” was replaced with “Figure 2”.

Reviewer 2 Report

The work presented in the paper continues the nice chemistry of Trofimov's school based on the use of super-base concept. Synthetic chemistry described here is of highest quality. It is really novel, and it will be interesting for a very broad community of scientists - from bio/medicine chemistry to different areas of coordination chemistry. The work is perfectly done, and results are well scientifically described. 

I would suggest to publish the paper as it is. Maybe just minor language checking is required.

Author Response

English was checked. 

Reviewer 3 Report

This manuscript describes a synthesis of 3,5-diaryl-2,6-dicyanoanilines by three-component condensation between acylethynylpyrroles and (as the authors show) two equivalents of malononitrile. If the pyrrole nitrogen is unprotected, its anion reacts in place of the second malononitrile. The products are carefully characterized (e.g., every structure has the right number of 13C lines) and most of the science is sound. The English is adequate though there are a few non-idiomatic phrases (e.g. "tarring being accompanied the process"). 

I don't follow the argument for (Z) stereochemistry in the C=C bond of aminopyrrolizines 6 (Figure 2). HMBC correlations are not evidence of stereochemistry, and the NOESY correlation between the vinylic H and the thienyl ring would be there regardless of C=C stereochemistry. If this point is important, I suggest the authors obtain a crystal structure. 

Author Response

Reviewer wrote:

I don't follow the argument for (Z) stereochemistry in the C=C bond of aminopyrrolizines 6 (Figure 2). HMBC correlations are not evidence of stereochemistry, and the NOESY correlation between the vinylic H and the thienyl ring would be there regardless of C=C stereochemistry. If this point is important, I suggest the authors obtain a crystal structure. 

Answer:

To confirm the Z-form of aminopyrrolizine 6c, 2M NMR spectra with appropriate explanations were added to the SI file.

The following sentences have been added to the manuscript:

“2M NMR spectra of aminopyrrolizine 6c confirm its  the Z-form”.